# Senescence-Associated Cell Transition and Interaction (SACTAI): A Proposed Mechanism for Tissue Aging, Repair, and Degeneration

**DOI:** 10.3390/cells11071089

**Published:** 2022-03-24

**Authors:** Yajun Liu, Jonah Schwam, Qian Chen

**Affiliations:** Laboratory of Molecular Biology and Nanomedicine, Department of Orthopaedics, Alpert Medical School of Brown University, Rhode Island Hospital, Providence, RI 02903, USA; Yajun_Liu@Brown.edu (Y.L.); Jonah_Schwam@Brown.edu (J.S.)

**Keywords:** cell senescence, SACTAI, SASP, aging-related disease, mesenchymal stem cell

## Abstract

Aging is a broad process that occurs as a time-dependent functional decline and tissue degeneration in living organisms. On a smaller scale, aging also exists within organs, tissues, and cells. As the smallest functional unit in living organisms, cells “age” by reaching senescence where proliferation stops. Such cellular senescence is achieved through replicative stress, telomere erosion and stem cell exhaustion. It has been shown that cellular senescence is key to tissue degradation and cell death in aging-related diseases (ARD). However, senescent cells constitute only a small percentage of total cells in the body, and they are resistant to death during aging. This suggests that ARD may involve interaction of senescent cells with non-senescent cells, resulting in senescence-triggered death of non-senescent somatic cells and tissue degeneration in aging organs. Here, based on recent research evidence from our laboratory and others, we propose a mechanism—Senescence-Associated Cell Transition and Interaction (SACTAI)—to explain how cell heterogeneity arises during aging and how the interaction between somatic cells and senescent cells, some of which are derived from aging somatic cells, results in cell death and tissue degeneration.

## 1. Introduction

Aging is often defined as progressive physiological degeneration, leading to impaired function and increased vulnerability to death. It has become an increasingly burdensome challenge due to the rapid growth of the aging human population. A 2015 report from the UN estimates that by 2050, there will be more than 2.1 billion people over the age of 60 due to an increased birth rate, improved healthcare, and better living conditions [1]. Several diseases are strongly correlated to advanced age, including Alzheimer’s disease, cardiovascular disease, osteoarthritis, diabetes and cancer; all roughly double in incidence every five years after age 60 [2,3]. So far, many ARDs lack effective cures [4]. Although recent studies have been conducted to evaluate the biological sources of aging, the biological responses and compensations to aging, and the possibility to medically delay aging, there are still numerous challenges to understand this complex biological process.

The decline in organ and tissue function of ARD reflects the underlying molecular aberrations driving the cellular aging process [2]. Nine causative mechanisms of aging have been proposed with the criteria that each occurs during normal aging, accelerates aging when experimentally aggravated, and decelerates aging when experimentally ameliorated. These hallmarks include cellular senescence, stem cell exhaustion, altered intercellular communication, genomic instability, telomere attrition, epigenetic alterations, loss of proteostasis, deregulated nutrient sensing, and mitochondrial dysfunction [4]. Cellular senescence is not only a leading cause of aging, but also the cellular basis of the other causative mechanisms of aging. Although remarkable progress has been made in understanding cell senescence since it was proposed by Hayflick sixty years ago [5], it is still a work in progress and a hot area of research currently. Still unclear is how cell senescence contributes to the major phenotypes of ARDs including fibrosis, soft tissue ossification (calcification), inflammaging (chronic low-grade inflammation), cell death and tissue degeneration. Uncovering the molecular and cellular mechanisms of cell senescence will have a profound impact on our understanding of aging and developing interventions of ARDs.

Aged tissues and organs contain a complex mixture of quiescent, proliferative, and senescent cell populations [6]. The cell numbers, types and functions change dynamically during aging, which are often thought to contribute to disease progression. Cellular senescence is closely associated with such processes. How a small pool of senescent cells leads to tissue degeneration remains to be unveiled. It is of great importance to understand the roles of different cell types, and how they change and interact with each other spatially and temporally during aging. In this review, we will start with the recently updated understanding of cell senescence and death, discuss the different cell types in tissues, milestones of cellular senescence, and their roles in regeneration and degeneration, and conclude with the implication of the newly proposed mechanism—senescence-associated cell transition and interaction (SACTAI) in ARDs.

## 2. Cell Senescence Is Cell-Autonomous: Telomere Shortening, Replicative Stress and SASP Manifestation 

Almost 60 years ago, Leonard Hayflick and Paul Moorhead first reported that primary human cells derived from embryonic tissues exhibited limitations while dividing [5]. These fibroblasts could only divide for 40–60 times in cell culture before entering senescence. Hayflick defined the different stages of cell culture into three phases (Figure 1). Phase I is the original primary culture, where cells divide to cover the surface at a relatively slow rate. Phase II represents the period when robust cell division occurs. Continuous subcultures of the same batch of cells leads to Phase III, when cells not only stopped growth but also died in a short period of time. Hayflick postulated that cell alterations must have occurred during cell growth in Phase II, which led to cell growth arrest and death in Phase III. He called this phenomenon cell senescence and further postulated that it could be a mechanism to counter unlimited cell proliferation in cancer. This phenomenon of cell growth arrest is later defined as Hayflick Limit due to replicative senescence (Figure 1A). In a discovery that led to a Nobel prize on telomerase, it was found that the primary cause for Hayflick Limit, or replicative senescence, is telomere shortening [7,8].

Telomeres are non-coding regions at the end of chromosomes, which consist of thousands of the same sequence repeats [7,8]. Due to the nature of lagging-strand synthesis, DNA polymerase cannot replicate the full 3′ end of duplex DNA [9]. This leads to the loss of small segments of DNA within telomeres. After a certain number of divisions, the telomeres reach a critical length where cells become senescent. Thus, telomere shortening drives the activation of cell senescence mechanisms. However, cell growth arrest by itself may not be harmful to our body, since most somatic cells in a mature adult organ are post-mitotic or divide very slowly. The detrimental effects of cell senescence were not clear until the discovery of the senescence-associated secretory phenotype (SASP) [10].

The discovery of SASP demonstrated that, even though they no longer proliferate, senescent cells remain metabolically active and secrete a broad spectrum of factors, including cytokines, chemokines, proteases, growth factors, receptors and ligands, and extracellular matrix components [10]. These SASP factors can cause autocrine and paracrine inflammatory responses. Some of the SASP factors, such as interleukin-1 (IL-1), IL-6, IL-8, TNF-α, and matrix metalloproteinase1 (MMP1) [11], contribute to persistent chronic inflammation in the microenvironment, which is often associated with multiple aging-related phenotypes [12,13]. In addition to SASP, senescent cells also influence the microenvironment through cell the secretome including microvesicles (MVs). Typically, MVs function as a key component of the cell secretome in immunomodulatory regulation and tumor growth inhibition [14,15]. However, senescent cells change the composition of MVs to negatively impact the microenvironment. Exosomes, an important type of MVs, contain biological active molecules including cytokines, growth factors, and regulatory miRNA [16]. Senescence greatly alters the composition of their exosomes, especially the miRNA content. RNAseq analysis of senescent mesenchymal stromal cell (MSC) MVs identified many highly expressed genes associated with ARDs [17]. The size and source of MSC MVs and exosomes have been reviewed previously [16].

## 3. Cell Senescence-Associated Apoptosis and Degeneration Is Senescent Cell Non-Autonomous: The SACTAI Mechanism for Aging

While the research in the past sixty years has established the cell senescence process that leads to the Hayflick limit, some questions remain unanswered in the original Hayflick cell culture experiment. The Hayflick phenomenon included Phase III when cell number rapidly declines after reaching the Hayflick Limit (cell senescence) (Figure 1A). While the mechanism for the increased cell number to result in cell senescence in Phase I and II has been elucidated, the mechanism for senescence-associated cell death in Phase III was not addressed or even ignored. Cell death (apoptosis) and degeneration are also hallmarks of ARDs in vivo. However, senescent cells are resistant to apoptosis, which is a characteristic of cell senescence [18,19]. The significant decrease in cell number in aging tissues in vivo and in primary cell cultures in vitro is likely not due to apoptosis of senescent cells, but other non-senescent cells. Thus, while cell senescence is a cell autonomous process, cell senescence-associated apoptosis and degeneration may not be. How does this cell senescence-associated apoptosis and degeneration occur?

Based on RNAseq of human aging cartilage tissues and primary chondrocyte culture passage experiments, we proposed a mechanism of Senescence-Associated Cell Transition and Interaction (SACTAI) [20]. SACTAI is induced by repeated activation of somatic cell (SomC) replication in response to stress signals during aging or after injury. This causes cell type transitions from differentiated SomCs to MSCs by de-differentiation, and further into senescent MSC (snMSC) due to replicative stress (Figure 1B). Although only a single type of SomCs, in this case chondrocytes, exists in Phase I, different cell types including MSCs and snMSCs are induced by cell replication in Phase II, and co-exist with SomCs (Figure 1B). Such SomC transitions result in heterotypic interactions between snMSC and non-senescent SomC, which lead to SomC death and total cell number decreases in Phase III. Thus, the cell alteration event postulated by Hayflick may contain replicative senescence-associated changes of cell phenotype from SomC to SnMSC (Figure 1B). While this process is cell autonomous, the subsequent cell death event is senescent cell non-autonomous. It involves interaction of snMSC with non-senescent SomC, probably through SASPs. The experimental evidence to support the SACTAI mechanism is presented in the following.

Although it was thought for a long time that chondrocytes were the only cell type in cartilage tissue, evidence for the existence of different cell types has been shown in aging human articular cartilage in recent years [21,22,23,24,25]. Further, the percentage of progenitor/MSC was increased in osteoarthritis (OA) cartilage, suggesting aging-induced cell heterogeneity [22,26]. Two laboratories, one in US and one in UK, successfully isolated and characterized CD166-positive cell lines from articular cartilage of OA patients [27,28]. The membrane glycoprotein ALCAM (CD166) is a unique progenitor/MSC marker, which is not expressed by chondrocytes. Two pools of MSC populations were identified by both laboratories independently. Both MSC pools were multi-potent, expressed low levels of chondrogenic marker type II collagen (COL2A1) and high levels of fibronectin receptor (CD149e), characteristic of MSCs. Comparison between the two pools of MSCs (Figure 2) indicated that the more proliferative (less senescent) MSC population had normal telomere length compared to normal cartilage stromal cells (NCSC), a similar level of aggrecan (ACAN) as chondrocytes and were prone to chondrogenesis. In contrast, the more senescent MSC population had reduced telomere length, lower levels of ACAN and increased senescent-associated β-galactosidase (SA-β-gal) activity. The senescent MSC population, which is termed OA-MSC, express higher levels of the MSC marker CD90 and are prone to osteogenesis [27,28]. RNAseq analysis indicated that OA chondrocytes (OAC) expressed the lowest level of p16, a cell senescence marker while NCSCs expressed higher levels of p16 and OA-MSC expressed the highest level of p16. This established the cell senescence pathway in cartilage from OAC to NCSC to OA-MSC. OA-MSC, the terminally senescent cells in cartilage, expressed high levels of SASPs including proinflammatory cytokines IL-1b, IL-6, IL-8, growth factor TGF-b1, and morphogen Sonic Hedgehog [20,29]. Interestingly, OAC expressed receptors for these signaling molecules [20]. Thus, senescent OA-MSCs are a source of SASPs while the somatic OACs are recipients of SASP signaling in OA cartilage.

A significant increase of MSCs in OA cartilage could be from outside or inside cartilage [22,26]. One possibility is that MSCs come from outside sources such as blood vessels or synovial fluid, which were transported into the cartilage. However, the lack of blood vessels and the dense cartilage matrix in cartilage make MSC infiltration from outside difficult if not impossible [30]. Another possibility is from an inside cell source, i.e., chondrocytes, through cell transition. By conducting a serial primary cell culture passaging experiment like Hayflick’s, Liu et al. demonstrated a gradual and progressive conversion of a cell population containing 97.1% chondrocytes (CD166 negative) at passage zero to a population containing 98.6% MSC (CD166 positive) at passage five [20]. Cell morphology changed from cuboidal shape of chondrocytes in early passages (P0–P2) to the long spindle shape of fibroblasts in late passages (P3–P4). At P5, some cells are large with processes representing typical senescent cell morphology. Pro-inflammatory cytokine/chemokine CXCL1 and IL-8 were induced after P3. These data suggest that OACs were converted to senescent OA-MSCs through cell passaging. Such cell transition is driven by repeated replication, which leads to MSC senescence and death of OAC.

## 4. Two-Way Communications between Senescent and Non-Senescent Cells: Altered Intercellular Signaling Mechanisms during Aging

An important consequence of senescence-associated cell transition is enabling heterotypic cell interaction between SnMSCs and non-senescent SomC. SASPs appear to be major signaling molecules mediating the SACTAI, as shown by recent publications [29,31]. The interaction between senescent and non-senescent cells is a two-way communication. The first is signaling from senescent cells to non-senescent cells (Figure 3). snMSC acquire a pro-inflammatory phenotype and secrete cytokine IL-1β due to cell transition and senescence. On the other hand, chondrocytes are the main recipient cells of inflammatory signaling by expressing IL-1R [20]. In addition to cytokine signaling, a recent study also identified Sonic Hedgehog (SHH) to mediate senescence signaling from snMSC to SomC [29]. SHH is known to play an important role during skeletal development and homeostasis [32,33]. SHH is expressed by NCSCs and increased in OA-MSCs during aging [29]. The SHH receptors (PTCH1 and SMO) and transcription factors (GLI2 and GLI3) are expressed by chondrocytes. SHH autocrine treatment of MSC stimulated proliferation, chondrogenesis, hypertrophy, and replicative senescence through increasing SASPs, such as IL-1β, IL-6, CXCL1 and CXCL8. SHH paracrine treatment of OACs activated catabolic signaling in chondrocytes including suppressing anabolic ECM COL2A1 and increasing catabolic MMP13. Incubation of non-senescent OAC with the conditioned medium of senescent OA-MSC triggered massive cell death of OAC [29]. The senescence-triggered catabolism and apoptosis of non-senescent OAC depends on SHH, as SHH knock-down in OA-MSC abolishes its induction of chondrocyte death. Interestingly, SHH was previously thought to be anti-senescence because of its property as a mitogen [32,34]. However, SHH induction of proliferation also resulted in replicative senescence and SASP, which interacted with OACs and resulted in their catabolism and death. Thus, SHH mediates senescence-associated interaction through two mechanisms: (1) Increasing MSCs proliferation and activating replicative senescence, and (2) promoting OAC catabolism and apoptosis by creating an inflammatory microenvironment favorable for tissue degeneration during OA pathogenesis. The newly discovered role of SHH in triggering MSC senescence and chondrocyte apoptosis revealed a new mechanism for the involvement of hedgehog signaling in cartilage aging and degeneration [35]. Interestingly, the role of SHH in coordinating mesenchymal cell proliferation, senescence, chondrocyte death and cartilage degradation during aging mirrors its role in organizing limb morphogenesis during embryogenesis where cell senescence was also observed in limb development [32,36].

The second senescence associated signaling pathway is for SomC to modulate SnMSC (Figure 3). TGF-β has been shown to play an important role in this process [31]. Human chondrocytes synthesize TGFβs including TGFβ1, 2, and 3. TGF-β had different effects on OA-MSC and OAC. OA-MSC expressed the same level of bone morphogenetic protein (BMP)-Receptor-1A as OAC but only 1/12 of transforming growth factor beta (TGF-β)-Receptor-1. While TGF-β specifically activated SMAD2 and chondrogenesis in OAC, it also activated BMP signaling-associated SMAD1, hypertrophy, mineralization, and MMP-13 in OA-MSC. Thus, SomCs, in this case OAC, modulated SnMSC through TGF-β lateral signaling, which resulted in enhanced calcification and abnormal ossification [31]. In summary, the SACTAI mechanism suggests that senescence-associated cell transitions and interactions are closely linked. Cell type transitions during cell aging and senescence result in the interactions of different types of cells. Rather than a linear relation between SomCs, MSCs and SnMSCs, SACTAI provides explanations for multi-level regulations among different cell types through looped feedbacks during aging (Figure 3). We used primary human chondrocytes from aging OA patients as an example to illustrate senescence-associated cell heterogeneity and interaction during OA. In the following sections, we will examine whether the SACTAI mechanism is also applicable to other tissues during aging. 

## 5. SACTAI in Adult Tissues

In a typical tissue, the cells may be of the same type or of multiple types. The majority of cells are very distinctive and perform specific functions. These SomCs are mature and terminally differentiated cells, which usually carry out specialized tasks in the body. MSCs can divide and differentiate, are responsible for tissue morphogenesis during development and growth and repair during the adult stage. The MSCs in the adult stage are the focus here. Together SomCs, MSCs, and SnMSCs ensure the proper functions of an organ and maintain tissue homeostasis. 

### 5.1. Dedifferentiation: SomC to MSC Transition

In adult tissue, the most common cellular state is that of terminally differentiated SomCs. A plethora of different terminally differentiated SomCs comprises each adult tissue, with each cell type taking on a unique function within the tissue. These SomCs are often thought to have permanently exited the cell cycle and are incapable of replicating or replacing themselves [37]. Maintaining a stable cellular identity is crucial for normal tissue function. Such stability is achieved through epigenetic regulation [38]. However, cell identity can be altered under experimental conditions. For example, John Gurdon performed nuclear transplantation on mature adult cells and converted them into cells with properties of a fertilized egg [39]. Such experiments have found that these terminally differentiated SomCs have the capacity to de-differentiate. De-differentiation is defined as the reversion of a fully differentiated cell into one with stem/progenitor cell property [40]. The discovery that SomCs can be experimentally manipulated to become pluripotent suggest the possibility of cells changing their identity, also known as cellular plasticity [38].

Although de-differentiation can be forced experimentally, these processes may also occur physiologically due to tissue injury and/or loss of functional cells. It has been demonstrated that differentiated somatic cells of different tissue origins can dedifferentiate to mesenchymal stromal cells in response to injury [41,42]. These differentiated cells include both mesenchymal (e.g., muscle) and epithelial (e.g., intestine) in origins. The epithelial to mesenchymal transition (EMT) is a process by which epithelial cells lose their cell polarity and cell–cell interaction and gain the MSC properties of cell migration, ECM production, and capacity of differentiating into different cell lineages [43,44,45]. In addition to development, EMT also occurs during wound healing and aging including organ fibrosis and metastasis of cancer progression.

### 5.2. MSC Senescence

MSCs from development are a small population of cells that reside in the tissue throughout most postnatal life [46]. They are undifferentiated cells and give rise to a limited number of mature cell types that play roles in the functions of a tissue [38,47]. These adult stem cells are self-renewing, clonogenic, and multipotent in nature. Thus, they are critical to maintaining the tissue homeostasis. Adult stem cells are generally in a predominantly quiescent state for long-term maintenance of tissue functions [48,49]. Upon the loss of functional cells either through aging or injury to the tissue, they are activated to proliferate and differentiate into the required cell types. Therefore, adult stem cells play a key role in tissue repair. These MSCs are found in a variety of tissues, including bone marrow, adipose tissue, cartilage, synovial fluid, brain, skin, lung, dental pulp, umbilical cord blood, and placenta [50,51,52,53,54]. They can differentiate into multiple cell lineages, such as bone, cartilage, adipose, and neuronal cells [50]. Thus, MSCs have emerged as powerful tools for therapeutic applications in tissue engineering and regenerative medicine [55]. Applications of MSC-based therapy have been tested in different diseases, such as graft-vs.-host disease (GVHD), diabetes mellitus (DM), Crohn’s disease (CD), multiple sclerosis (MS), myocardial infarction (MI), etc. [55,56]. Despite its huge potential in the field of disease treatment, cellular senescence greatly restricts the development of MSC-based therapy. After ex-vivo expansion of MSCs, they reach a senescent phenotype due to stem cell exhaustion [57,58]. Accumulating data have demonstrated discrepancies in the effects of MSC-based therapy, likely due to senescence-induced alterations [55,59,60]. A phase III trial to treat GVHD using MSCs showed no significant difference compared to placebo [61]. Another example of using bone-marrow derived MSCs to treat ischemic heart failure also showed no significant differences between MSC treatment and placebo [62,63]. Therefore, it is critical to monitor the self-renewal property of MSCs and understand what triggers its senescence transition.

Besides normal MSCs, another type of MSCs exists in the same microenvironment and increases its cell numbers with aging, known as SnMSCs. They are still inherently MSCs due to the expression of MSC surface markers, such as CD90, CD73 and CD105 [64,65]. But they no longer retain the self-renewal and differentiation potentials of normal MSCs [66]. SnMSCs have unique features different to normal MSCs, such as an enlarged and more granular morphology, condensed heterochromatin structure in the nuclei, changes in epigenetic regulation, shortened telomeres, and changes in differentiation potentials [66,67,68,69]. Characteristics of cellular senescence also apply to SnMSCs, such as increased expressions of senescence markers SA-β-gal, p16^INK4A^ and p53 [70].

### 5.3. BM-MSC Senescence

Bone marrow-derived MSCs (BM-MSCs) are the most investigated MSCs. They play a role in the regulation of the local microvascular network, differentiation to osteoblasts, and maintain the hematopoietic microenvironment [71]. Multiple studies have shown cell type transition in BM-MSCs. Maderia et al. performed ex-vivo cultivation of primary human BM-MSCs from healthy donors [72]. When comparing an early passage P3 to a late passage P7, they found a loss in the proliferative, clonogenic and differentiation potential. The cell morphology was also altered with cells becoming bigger and with a more granular cytoplasm, all characteristics of senescence. Compared to the early passage, P7 cell proteome analysis showed decreased levels of chaperone proteins and stress response proteins (BiP, HSP27, and HSP70), and increased levels of apoptosis-related proteins (annexin and VDAC1), cell cycle regulation and aging-related proteins (eIF3F and PSMD11). These changes in protein expression profiles in the late passages suggest that cellular senescence occurred during ex-vivo expansion of BM-MSCs.

Similarly, Bertolo et al. cultured human primary BM-MSC in vitro for up to 11 passages [73]. At early passages, BM-MSCs were actively dividing and maintaining small and homogeneous shapes. In late passages, cells reached replicative senescence with a larger and asymmetric shape. This long-term ongoing culture showed increases in SA-β-Gal activity and senescence marker CDKN2A (p16) expression and decreases in telomere length. Interestingly, the expression of CDKN2A (p16) did not change much in the initial six passages. There is likely a threshold for MSCs to shift their balance from replication and differentiation to becoming senescent. Wagner el al. also cultured primary human BM-MSCs in vitro and observed similar changes in cell morphology, differentiation potentials, mRNA expression, and ultimately proliferation arrest [74]. These changes are not restricted to late passages but occurred continuously with increasing passages.

## 6. SACTAI: A Mechanism of Balancing Tissue Repair and Degeneration

In healthy tissues/organs, cellular senescence is not only a well-known hallmark of aging, but also an effective mechanism to prevent proliferation of damaged cells. Stem cell senescence is a cellular response to endogenous or exogenous stresses, including accumulated stresses through aging. During the early stages of life, it is a protective mechanism to limit proliferation and function of aberrant cells [75]. However, cell senescence is not always good, especially in aged tissues. The accumulation of senescent cells during aging contributes to disease pathogenesis and exhibits negative effects on non-senescent cells [75]. Such changes from beneficial to detrimental effects can be explained by SACTAI. In a young adult, the tissue environment contains mostly differentiated SomCs with specific roles to ensure proper function of the organ. However, a small pool of quiescent MSCs is left in the tissue as a backup for repair. When a young individual encounters a stress and requires damage repair, MSCs exit the quiescent state and resume their abilities for self-renewal and differentiation. Such robust cell division correlates with Hayflick’s Phase II before the SnMSC is induced (Figure 1). Thus, SomCs and MSCs maintain a balance to carry out tissue regeneration without reaching their replication capacity. In a young adult tissue, tissue repair is balanced with tissue degeneration.

However, the accumulation of stress events through aging leads to repeated activation of repair. Upon reaching replication capacity, some MSCs become SnMSCs. In order to maintain repair, SomCs de-differentiate into MSC-like cells with potentials to replicate and differentiate. This cell transition occurs with the intent to balance out the loss of normal MSCs but is often insufficient to rescue long term degeneration. In aged tissue, more and more MSCs transit into SnMSCs. Accumulated senescent cells increase secretion of inflammation-related factors such as SASPs to create an inflammatory microenvironment (Figure 3). SomCs become the targets of SASPs by expressing receptors for inflammation signals. A high inflammation environment triggers cytokine-induced catabolism and apoptosis of SomCs, correlating with the rapid decline of cell number in Hayflick’s Phase III (Figure 1B). In an aging tissue, tissue fibrosis and degeneration may be more than proper regeneration, contributing to organ failure and ARD progression.

As mentioned above, senescence-associated cell transition is similar to a well-studied cell transition pathway, epithelial to mesenchymal transition (EMT). EMT is also reversible, known as a mesenchymal-epithelial transition (MET) [43]. During embryonic development, the roles of EMT/MET are indispensable as they contribute to the formation and function of developing organs [44]. However, EMT/MET transitions in adult life often play roles in ARDs including organ fibrosis and cancer [76,77]. In liver fibrosis, activation of pro-EMT pathways, such as TGF-β, contributes to the source of myofibroblasts, which in turn secret pro-fibrotic factors [78,79]. In intestinal fibrosis, EMT is the main source of myofibroblasts, and contributes to ulcerative colitis and Crohn’s disease [80,81]. In ocular fibrosis, EMT is the only source for myofibroblasts [82]. Under TGF-β stimulation, keratocytes, lens epithelial cells, trabecular meshwork cells, and retinal pigment epithelial cells can undergo EMT to acquire myofibroblast phenotype [82]. SACT, like EMT, is involved in cell phenotype conversion from a specialized quiescent somatic cell to a mesenchymal stem cell. The SomC/MSC transition is also reversible and known as the de-differentiation/differentiation process, responsible for tissue repair and regeneration (Figure 3). It also occurs not only during development but also in the pathological processes during adulthood, representing both regeneration and degeneration at different stages.

Direct serial passaging of cell lines has shown the preference to push the cell transition towards senescent cell phenotypes. Although senescent cells have been defined as irreversible, some evidence suggests that senescence may not be the end of the road. There may be still a potential for them to be rejuvenated. Aged skeletal stem/progenitor cells (SSPCs) from 52-wk-old wildtype mice treated with sodium salicylate, a low-grade anti-inflammatory agent to inhibit NF-κB, showed decreased senescence, increased cell number, and increased osteogenic function [83]. Senescent retinal pigment epithelium (RPE) cells, when co-cultured with embryonic stem cells, regained the proliferative capacity and suppressed senescence markers (SA-β-gal, p16^INK4A^, p21^Cdkn1a^ and p53) [84]. However, whether those cell type transitions can occur physiologically is unknown.

## 7. SACTAI in Age-Related Diseases

ARDs are triggered by the stress signals that result in cell senescence including oxidative, irradiation, chemical, mechanical, inflammatory, and replicative stress [75]. The concept of SACTAI can be applied to cells in different tissues during aging. Both in vivo and in vitro data have indicated the cell type transitions and interactions in different tissues, and how they contribute to ARDs (Figure 3). It provides a general mechanism for aging and aging-associated tissue repair and degeneration.

### 7.1. Physical Dysfunction

Physical function declines with aging, causing disability, increased health expenditure and mortality [85,86]. Obesity and physical inactivity are often associated with physical dysfunction [87,88]. The cellular pathogenesis of age-related physical dysfunction is not fully understood. There are currently no mechanism-based interventions for improving physical function in the elderly. Adipose tissue is the major source of obesity and inactivity-related inflammation [87]. It contains a population of MSCs with multilineage differentiation along adipogenic, osteogenic, chondrogenic, and neurogenic lineages. Wall et al. used human adipose-derived MSCs (AD-MSCs) for serial passaging [89]. The growth rate was stable through P5 and decreased significantly afterwards. Although AD-MSCs were capable of both adipogenic and osteogenic differentiation, osteogenic differentiation started to dominate at later passages. There were also changes in mRNA expression of osteogenic and adipogenic markers, and production of calcium deposits and lipid vacuoles. By P6, AD-MSCs in adipogenic medium had significant decreases in lipoprotein lipase and PPARϒ. Similarly, by P6, AD-MSCs in osteogenic medium also had a decrease in alkaline phosphatase expression. Reduced growth rate and changes in differentiation potentials are indicators of cell transition into senescent AD-MSCs. Kirkland’s group provided more evidence for AD-MSCs and their senescence-associated cell interactions by transplanting senescent adipose-derived MSCs into young mice [86]. Characteristics of physical dysfunction were observed in these mice, including reduced walking speed, muscle strength and physical endurance. These transplanted senescent cells had a limited survival time. However, senescence persisted long-term due to cell transition caused by the transplanted senescent cells. The visceral adipose tissue had increased SA-β-gal activity and p16^INK4A^ expression, indicating transition and interaction between senescent cells and normal cells. 

### 7.2. Cardiovascular Disease

Cardiovascular disease is the most common cause of death in elderly people. It includes chronic ischemic heart disease, congestive heart failure and arrhythmia [3]. Adult human heart is known to harbor a small population of resident stem cells, also known as cardiac stem and progenitor cells (CPCs) [90,91,92]. These CPCs enable the adult heart to self-renew over the human lifespan [93]. The function of CPCs decreases with aging, leading to cardiovascular disease [94,95]. Ellison-Hughes’ group showed that senescent CPCs secreted SASP factors that negatively affected normal CPCs and caused cell transition into senescent CPCs [96]. Human CPCs were isolated from cardiac surgery patients with cardiovascular disease (aortic disease, valve disease, coronary artery bypass graft or multiple disease). Analysis showed an increase of senescent CPCs in old patients, with more than half of CPCs being senescent. These senescent CPCs have diminished self-renewal, differentiation and regenerative potentials. They also secreted high levels of SASPs, including IL-1β, IL-6, IL-8, MMP-3, PAI1, and GM-CSF. When normal CPCs were incubated with conditioned medium from senescent CPCs, there was a decrease in cell proliferation and an increase in senescent markers (SA-β-gal and p16^INK4A^). In vivo interaction was also tested by transplanting senescent CPCs into young mice. Data showed that senescent CPC injection leads to heart failure, because of the diminished regenerative and repair capacity of senescent CPCs.

Vascular calcification contributes to coronary artery disease and peripheral artery diseases and is closely associated with cardiovascular events and mortality [97,98]. Nakano-Kurimoto et al. cultured vascular smooth muscle cells (VSMCs) from human coronary artery and compared early (P6) and late passages (P13) [99]. Cells in late passage became senescent with reduced proliferative activity, flat and enlarged morphology, and increased expression of senescence markers including SA-β-gal, p16^Ink4a^ and p21^Waf1/Cip1^. Senescent VSMCs were also more susceptible to calcification and showed enhance expressions of genes in osteoblasts. These data suggest the VSMCs also undergo osteoblast transition during senescence.

### 7.3. Neurodegenerative Disorder

Neurodegenerative disorders are characterized by progressive neuronal death and loss of specific neuronal cell populations. Axonal degeneration takes place as a consequence of normal aging. Thus, age constitutes the main risk factor for its pathogenesis. Neurodegenerative disorders include Alzheimer’s disease, Parkinson’s disease, multiple sclerosis (MS), Huntington’s disease, and amyotrophic lateral sclerosis [100,101]. It has been shown that neural progenitor cells (NPCs) play important roles in neurodegenerative disorders. NPCs have the capacity to differentiate into neurons, astrocytes and oligodendrocytes to replace the damaged cells in adult brain [101]. Croker’s group identified senescent NPCs from primary progressive multiple sclerosis (PPMS) [102]. These senescent NPCs were located within demyelinated white matter lesions. They expressed high levels of senescence markers, such as SA-β-gal, p16^INK4A^ and p53. An extracellular protein associated with senescence HMGB1 was also secreted by senescent NPCs. HMGB1 can change gene expression profiles and impair the maturation of oligodendrocyte progenitor cells (OPCs), which differentiate into myelinating oligodendrocytes for tissue regeneration [103,104]. When incubated with conditioned medium from senescent NPCs, OPCs showed increases in multiple senescence markers, including MMP2, p16^INK4A^, and IGFBP2. Therefore, senescent NPCs can cause senescence associated cell transition of OPCs through indirect interactions by secretion of senescence markers.

### 7.4. Idiopathic Pulmonary Fibrosis

Idiopathic pulmonary fibrosis (IPF) is a chronic interstitial lung disease defined by a progressive and irreversible loss of lung function through scar tissue accumulation [105,106,107]. There is an increase in cell senescence markers in lung fibroblasts from IPF patients [107,108,109,110]. Rojas group found that BM-MSCs from IPF patients were senescent with mitochondrial dysfunction and accumulation of DNA damage [111]. When normal human lung fibroblasts were cultured with conditioned media from these senescent BM-MSC, they showed an increase in expression of senescence markers, such as SA-β-gal, p16^INK4A^ and p53. These results suggest that senescent MSCs could act on normal MSCs through the cell secretome and induce their senescence. Thus, these senescent cells contribute to the pathogenesis of the disease.

### 7.5. Bone and Joint Degeneration

OA is a common age-associated disease in the knee joint. It is a consequence of progressive age-induced senescence in the cartilage, leading to changes such as articular cartilage degradation, osteophyte formation, and subchondral bone sclerosis [112,113,114]. Repair of the joint cartilage relies on chondrocytes and resident MSCs. These MSCs are often found in articular cartilage, sub-chondral bone marrow and synovial tissue [115,116,117,118]. With cell transitions toward senescence, BM-MSCs also showed senescence-associated interactions. Brondello’s group co-cultured human BM-MSCs with OA chondrocytes for 7 days in a without-contact system [119]. Proliferative MSCs significantly reduced the expression of hypertrophic markers (TGF-β1, ADAMTS3 and ADAMTS5) associated with SASP, the fibrotic marker collagen 3 (Col3) and senescence associated cell cycle inhibitors (p16^INK4a^, p15^INK4b^ and p27^KIP1^) in OA chondrocytes. Senescent MSCs retained the anti-fibrotic property by repressing Col3, but no longer suppressed p16^INK4a^, p15^INK4b^, p27^KIP1^, TGF-β1, and ADAMTS3 in OA chondrocytes. Therefore, senescent MSCs participate in chondrocytes’ loss of function through cell interactions. In vivo interaction was also tested using BM-MSCs isolated from mice. These BM-MSCs were cultured until they reached the senescence state with upregulated p16^INK4a^, p19^ARF^, p21^Cdkn1a^, TGF-β1, and MMP-13. They also showed a significant increase in SA-β-gal activity. These senescent MSCs were then injected intra-articularly into wild-type mice, and sufficiently induced cartilage degeneration.

Periostitis is a condition where the periosteum surrounding the bones is under chronic or acute inflammation [120]. Periosteum is required for bone development and remodeling, and thus is essential for fracture healing of the bone [121]. It carries out postnatal fracture repair through a set of MSCs residing in the periosteum [122,123]. Vozzi et al. used periosteum-derived progenitor cells (PDPCs) harvested from human periosteal tissue for serial passaging [124]. At early passages (P3–P6), cells exhibited a low level of SA-β-gal activity with an elongated morphology in most cells. Starting from P12, PDPCs showed increased SA-β-gal activity with a much larger and irregular morphology. Cell proliferation marker Ki67 decreased gradually from P3 to P24. Senescence marker p16 increased from P3 to P24. These results suggest that PDPCs undergo senescence-associated cell transition.

## 8. Conclusions

### 8.1. Recent Finding

Recent genomic analysis reveals a remarkable heterogeneity of cell types during aging. Such cell heterogeneity gives rise to not only senescent cells but also other types of cells including progenitor and stromal cells. Adult MSCs constitute a small percentage of cells responsible for repair upon tissue damage. The increase in senescent cells is tightly associated with repeated activation of adult MSCs, where they reach replication capacity and become senescent. In response to stress signals, differentiated SomCs may lose their identity and de-differentiate into MSC-like cells for repair. Such epigenetically re-programmed MSCs are subject to cell senescence triggered by replicative, mechanical, and inflammatory stress signals. Although in small numbers, senescent MSCs manifest SASP, spread inflammation, and signal surrounding somatic cells in the tissue microenvironment. Thus, senescent MSCs may accumulate during aging by cell proliferation, transition, and senescence, and accelerate catabolism and death of somatic cells through cell interactions. Important signaling molecules mediating the SACTAI process include pro-inflammatory cytokine IL-1β, IL-6, IL-8, growth factor TGF-β, and morphogen Sonic Hedgehog, which at least partially overlap with SASPs.

### 8.2. Summary

SACTAI is a proposed two-step mechanism for aging-associated tissue degeneration and somatic cell death. In the first step, a few adult SomCs, in response to mechanical, inflammatory, or replicative stress signals, undergo proliferation, MSC transition and senescence, resulting in senescent MSCs (snMSCs). This cell transition and senescence process results in a heterogenous cell population, which enables heterotypic cell interactions with each other. During the second step, snMSCs interacts with SomCs via SASPs. Such cell senescence-associated signaling contributes to cell death and tissue degeneration in ARD. The newly discovered SomC transition to snMSC during aging may explain the fibrosis, abnormal ossification (calcification), and inflammaging phenotypes often associated with aging tissues. The identification of the multi-step mechanism of SACTAI provides an opportunity to develop potential drugs to intervene during different stages of ARD pathogenesis.

### 8.3. Implication

SACTAI provides a general mechanism for aging in different types of tissues. With increasing adult stromal cells during aging-associated cell heterogeneity, cell type transition is crucial to understanding the cellular processes leading to cell senescence, fibrosis, inflammation, and degradation. It is a dynamic and gradual process based on the balance between regeneration and degeneration. As different cell types (SomCs, MSCs, and SnMSCs) co-exist in the same microenvironment, cell transition and interaction occur both spatially and temporally. The mesenchymal cell transition and senescence may not only account for fibrosis, abnormal ossification, and inflammaging phenotypes of ARDs, but also provide key intervention points for treatment. Future studies on the mechanism of SACTAI in different tissues will shed light on new interventions and treatments of ARDs.

## Figures and Tables

**Figure 1 cells-11-01089-f001:**
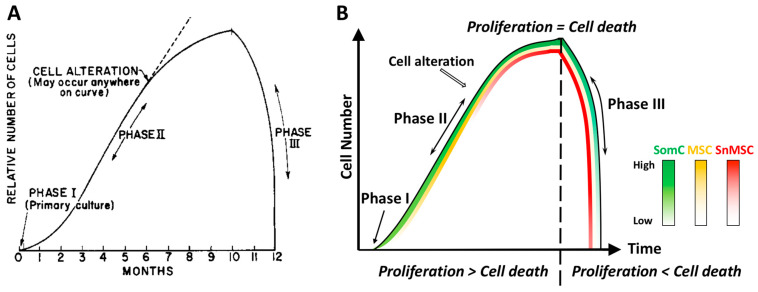
Diagram representations of the Hayflick phenomenon. (**A**) Classic Hayflick model [5]. Phase I is the primary culture. Phase II is defined as the multiple cell passage period characterized by robust cell proliferation and increase of cell number. After subcultures of 40–60 passages, cells reach Phase III with halting proliferation followed by rapid cell death. Such an event is termed Hayflick Limit, or replicative senescence. (**B**) The SACTAI model. Phase I contains SomCs. During Phase II, cell passaging-induced cell proliferation triggers cell alteration. Some SomCs are converted into MSC and SnMSC due to replicative senescence. Cell proliferation is more than cell death in Phase II. The emergence of SnMSC results in its communication with SomC via SASP, which triggers catabolism and death of SomC. At the end of Phase II, cell proliferation equals death resulting in a cell number plateau. During Phase III, SASP triggered SomC death results in a rapid decline of total cell number. The loss of SomCs depletes the cell pool for proliferation. As a result, SnMSC triggered SomC cell death outpaces its proliferation, resulting in tissue degeneration. Thus, different cell types (SomC, MSC, and SnMSC) contribute to the cell transition and interaction during various phases. They account for aging-associated cellular changes. SomC: somatic cell; MSC: mesenchymal stromal cell; SnMSC: senescent mesenchymal stromal cell.

**Figure 2 cells-11-01089-f002:**
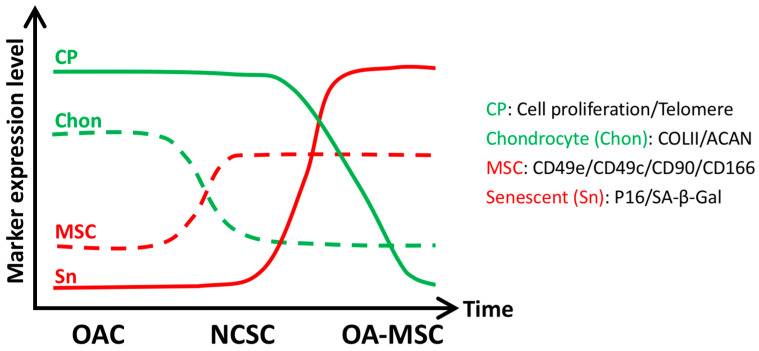
Comparisons of key characteristics among OACs, NCSCs, and OA-MSCs (Y-axis, marker expression level) during the aging process (X-axis, Time) [21,28]. Somatic cell markers including chondrocyte proliferation (telomere length) and differentiation (COLII/ACAN mRNA levels) decrease as cells transition from OAC to NCSC to OA-MSC during aging. Markers for cell senescence (P16/SA-β-gal) and MSC (CD49e/CD49c/CD90/CD166) increase as cells transition from OAC to NCSC to OA-MSC during aging. OA: Osteoarthritis; OAC: osteoarthritic chondrocytes; NCSC: normal cartilage stromal cell; OA-MSC: osteoarthritic-cartilage mesenchymal stromal cell.

**Figure 3 cells-11-01089-f003:**
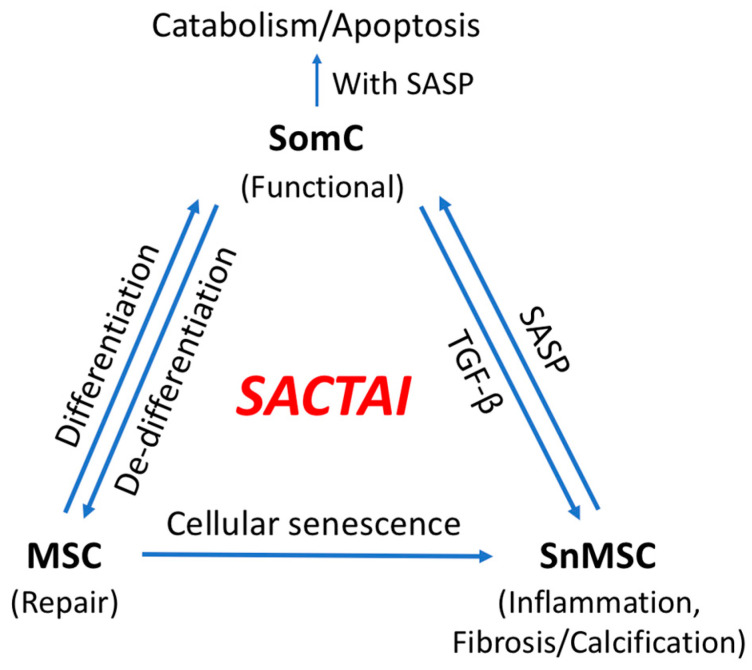
Senescence-associated cell transition and interaction (SACTAI) showing the relationship among SomC, MSC, and SnMSC. Due to aging-induced stress, MSCs proliferate and differentiate while SomCs de-differentiate and result in MSC-like cells for tissue repair. Repeated activation of MSC’s replication induces replicative senescence, resulting in SnMSCs. SnMSCs synthesize pro-inflammatory SASPs, which in turns act on SomCs to trigger catabolism and SomC death. Feedback of SomC with TGF-b reinforces the SnMSC properties including inflammation, fibrosis and abnormal calcification.

## Data Availability

Not applicable.

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
