# Peer review of "Senescence-Associated Cell Transition and Interaction (SACTAI): A Proposed Mechanism for Tissue Aging, Repair, and Degeneration"

_cells, 2022, doi:10.3390/cells11071089_

Round 1

Reviewer 1 Report

Thank you for the opportunity to review this manuscript by Liu and colleagues.  This Review article addresses cellular senescence, with a focus on the concept termed SACTAI.  The review is well-written, and provides the reader important insights into a relatively novel concept on the interplay between cell types in the tissue environment.  The use of Hayflick's original paradigm is quite effective in describing various phenomena.

The Review is highly focused upon a few mesenchymal cell types (e.g., cartilage, bone, heart), which is fine.  To include other organ types, especially those of organ systems that also undergo environmental stressors (e.g., skin, lung) and how senescence can lead to carcinogenesis would result in a less focused work.

The three figures, though simple, are very effective.

I only have a few minor suggestions to improve this superb work.

1).  Microvesicles and exosomes should be defined (size, source, etc--provide a generalized review) for those readers who are not familiar with these subcellular particles.

2).  Line 116-117 is unclear as written-   "Senescent cells also secrete SASP to influence the microenvironment through cell secretome, such as microvesicles (MVs)."   Unclear as to the purpose of this statement--is it that SASP are found solely in MVs?  

Author Response

We thank the reviewers for their constructive comments. We made revisions as suggested by the reviewers. All the revisions were marked for your convenience.  Our point-by-point response is as follows.

Reviewer 1:

Thank you for the opportunity to review this manuscript by Liu and colleagues.  This Review article addresses cellular senescence, with a focus on the concept termed SACTAI.  The review is well-written, and provides the reader important insights into a relatively novel concept on the interplay between cell types in the tissue environment.  The use of Hayflick's original paradigm is quite effective in describing various phenomena.

The Review is highly focused upon a few mesenchymal cell types (e.g., cartilage, bone, heart), which is fine.  To include other organ types, especially those of organ systems that also undergo environmental stressors (e.g., skin, lung) and how senescence can lead to carcinogenesis would result in a less focused work.

The three figures, though simple, are very effective.

I only have a few minor suggestions to improve this superb work.

1).  Microvesicles and exosomes should be defined (size, source, etc--provide a generalized review) for those readers who are not familiar with these subcellular particles.

Response: Thank you for your suggestion. We added a citation of a generalized review of microvesicles and exosomes for readers who are interested in their characteristics including size and source (line 133).

2).  Line 116-117 is unclear as written-   "Senescent cells also secrete SASP to influence the microenvironment through cell secretome, such as microvesicles (MVs)."   Unclear as to the purpose of this statement--is it that SASP are found solely in MVs?  

Response: We clarify the meaning of the sentence by stating that “In addition to SASP, senescent cells also influence the microenvironment through cell secretome including microvesicles (MVs).” (line 125)

Reviewer 2 Report

In the current review article, the authors describe a novel mechanism called Senescence-Associated Cell Transition and Interaction (SACTAI), to explain how cell heterogeneity arises during aging. Also, the mechanism of the interaction between somatic cells and senescent cells, as well as how some of the aging somatic cells lead to cell death and degeneration. In my opinion this is a well written review and a very interesting topic. All of my minor criticisms are merely editorial in nature.

The sentence in lines 46-47 on page 2 “Although remarkable progresses have been made in understanding of cell senescence since it was proposed by Hayflick sixty years ago [5], it..” should be “Although remarkable progress has been made in understanding of cell senescence since it was proposed by Hayflick sixty years ago [5], it..”

The sentence in lines 57-58 on page 2 “How a small pool of senescent cells lead to tissue degeneration remains to be unveiled.” should be “How a small pool of senescent cells leads to tissue degeneration remains to be unveiled.”

The sentence in lines 277-278 on page 8 “A plethora of different terminally differentiated SomCs comprises each adult tissue, with each cell type taking on a unique function within the tissue.“ should be “A plethora of different terminally-differentiated SomCs comprises each adult tissue, with each cell type taking on a unique function within the tissue.“

In Figure 2 legend please define OA, OAC, MSC, NCSC.

There are too many abbreviations throughout the whole manuscript that should be spelled out for people not expert in the field.

Some examples of in vitro are not in italics and should be corrected.

Author Response

We thank the reviewers for their constructive comments. We made revisions as suggested by the reviewers. All the revisions were marked for your convenience.  Our point-by-point response is as follows.

Reviewer 2:

In the current review article, the authors describe a novel mechanism called Senescence-Associated Cell Transition and Interaction (SACTAI), to explain how cell heterogeneity arises during aging. Also, the mechanism of the interaction between somatic cells and senescent cells, as well as how some of the aging somatic cells lead to cell death and degeneration. In my opinion this is a well written review and a very interesting topic. All of my minor criticisms are merely editorial in nature.

The sentence in lines 46-47 on page 2 “Although remarkable progresses have been made in understanding of cell senescence since it was proposed by Hayflick sixty years ago [5], it..” should be “Although remarkable progress has been made in understanding of cell senescence since it was proposed by Hayflick sixty years ago [5], it..”

Response: Changes have been made as suggested.

The sentence in lines 57-58 on page 2 “How a small pool of senescent cells lead to tissue degeneration remains to be unveiled.” should be “How a small pool of senescent cells leads to tissue degeneration remains to be unveiled.”

Response: Changes have been made as suggested.

The sentence in lines 277-278 on page 8 “A plethora of different terminally differentiated SomCs comprises each adult tissue, with each cell type taking on a unique function within the tissue.“ should be “A plethora of different terminally-differentiated SomCs comprises each adult tissue, with each cell type taking on a unique function within the tissue.“

Response: Changes have been made as suggested.

In Figure 2 legend please define OA, OAC, MSC, NCSC.

Response: They have been defined in Figure 2 legend.

There are too many abbreviations throughout the whole manuscript that should be spelled out for people not expert in the field.

Response: As suggested, we added a list of abbreviations next to the Keywords section.

Some examples of in vitro are not in italics and should be corrected.

Response: They have been corrected.